# Hydrogen Sulfide and Oxygen Homeostasis in Atherosclerosis: A Systematic Review from Molecular Biology to Therapeutic Perspectives

**DOI:** 10.3390/ijms24098376

**Published:** 2023-05-06

**Authors:** Constantin Munteanu

**Affiliations:** 1Faculty of Medical Bioengineering, University of Medicine and Pharmacy “Grigore T. Popa” Iași, 700454 Iași, Romania; constantin.munteanu.biolog@umfiasi.ro; 2Teaching Emergency Hospital “Bagdasar-Arseni” (TEHBA), 041915 Bucharest, Romania

**Keywords:** hydrogen sulfide, tissue oxygenation, hypoxia, atherosclerosis, homeostatic imbalances

## Abstract

Atherosclerosis is a complex pathological condition marked by the accumulation of lipids in the arterial wall, leading to the development of plaques that can eventually rupture and cause thrombotic events. In recent years, hydrogen sulfide (H_2_S) has emerged as a key mediator of cardiovascular homeostasis, with potential therapeutic applications in atherosclerosis. This systematic review highlights the importance of understanding the complex interplay between H_2_S, oxygen homeostasis, and atherosclerosis and suggests that targeting H_2_S signaling pathways may offer new avenues for treating and preventing this condition. Oxygen homeostasis is a critical aspect of cardiovascular health, and disruption of this balance can contribute to the development and progression of atherosclerosis. Recent studies have demonstrated that H_2_S plays an important role in maintaining oxygen homeostasis by regulating the function of oxygen-sensing enzymes and transcription factors in vascular cells. H_2_S has been shown to modulate endothelial nitric oxide synthase (eNOS) activity, which plays a key role in regulating vascular tone and oxygen delivery to tissues. The comprehensive analysis of the current understanding of H_2_S in atherosclerosis can pave the way for future research and the development of new therapeutic strategies for this debilitating condition. PROSPERO ID: 417150.

## 1. Introduction

Atherosclerosis is a multifactorial disease involving various cellular and molecular processes, including lipid metabolism, inflammation [1], mitochondrial dysfunction, autophagy, apoptosis, and epigenetics [2]. These processes can induce oxidative stress, which is characterized by an imbalance between oxidants and antioxidants in the body [3], leading to the generation of reactive oxygen species (ROS), reactive nitrogen species (RNS), and other free radicals that can cause damage to cellular components, including proteins, lipids, and DNA [4]. Mitochondrial dysfunction plays a crucial role in atherosclerosis as mitochondria are the primary source of ROS production [5]. These mechanisms contribute to the development and progression of atherosclerosis, characterized by the accumulation of lipids, calcium, fibrin, and cellular waste products in the arteries’ walls, leading to plaque formation [6]. Plaques can restrict blood flow and cause cardiovascular complications, such as heart attack, stroke, and peripheral artery disease. Atherosclerosis is a significant cause of morbidity and mortality worldwide, and multiple risk factors, including obesity, diabetes, smoking, high blood pressure, high cholesterol levels, and genetic predisposition, influence its development [7]. To fully understand the pathophysiological frame of atherosclerosis, it is necessary to consider the complex interplay of these cellular and molecular processes and how they contribute to the development and progression of this disease. Therefore, understanding the molecular mechanisms underlying atherosclerosis is critical for achieving effective preventive and therapeutic strategies [8].

Various cellular and molecular mechanisms contribute to the development and progression of atherosclerosis, including compromised tissue oxygenation, dysregulated autophagy, apoptosis, and epigenetic modifications, processes interconnected in a vicious cycle. Atherosclerosis compromits tissue oxygenation by reducing blood flow, which decreases oxygen delivery and can cause tissue hypoxia [9]. Autophagy is a physiological process involving cellular components’ breakdown and recycling [10]. Dysregulated autophagy can accumulate damaged proteins and organelles, contributing to oxidative stress and atherosclerosis [11]. Apoptosis, or programmed cell death, is another mechanism that can contribute to oxidative stress and atherosclerosis [12]. When cells undergo apoptosis, they release their contents, including ROS and proinflammatory mediators, which can further exacerbate the inflammatory response and promote atherosclerosis [13]. Epigenetic modifications, such as histone acetylation and DNA methylation, can also play a role in atherosclerosis by regulating gene expression [14]. Conversely, dysregulated epigenetic changes can lead to increased oxidative stress and inflammation, which influence the progression of atherosclerosis [15].

One of the critical biochemical events in the development of atherosclerosis is the oxidation of low-density lipoprotein (LDL) cholesterol [16]. Oxidized LDL is taken up by macrophages in the arterial wall, forming foam cells that contribute to developing plaques [17]. In addition, the accumulation of LDL and other lipids can activate proinflammatory pathways and release cytokines, chemokines, and other inflammatory mediators that further contribute to plaque formation [18]. Another critical aspect of the biochemical frame of atherosclerosis is the role of endothelial dysfunction, which is essential in regulating vascular tone and blood flow [19]. Dysfunction of the endothelium can be caused by factors such as inflammation, oxidative stress, and high blood pressure, leading to the development of atherosclerosis by promoting the accumulation of lipids and other inflammatory cells in the arterial wall [20]. Other factors that may contribute to the development of atherosclerosis include genetic predisposition and systemic inflammation [21].

Inflammation plays a critical role in atherosclerosis and is responsible for recruiting immune cells to the site of injury, where they release cytokines and other mediators that further exacerbate the inflammatory response [2]. Chronic low-grade inflammation leads to the recruitment of inflammatory cells, such as monocytes, macrophages, and T cells, to the site of vascular injury [22]. These cells release proinflammatory cytokines and chemokines that exacerbate the inflammatory response and promote the accumulation of lipids in the vessel wall [23]. The resulting lipid-laden macrophages, or foam cells, initiate the formation of atherosclerotic plaques. Inflammatory processes also contribute to plaque instability, which increases the risk of plaque rupture and subsequent thrombosis. Therefore, targeting inflammation has become a promising strategy for preventing and treating atherosclerosis [17].

The treatment for atherosclerosis depends on the severity of the condition and the individual patient’s risk factors. The main treatment objectives include minimizing the chances of developing complications, such as heart attack and stroke, and halting or decelerating the advancement of the disease [24]. Lifestyle changes are the first line of treatment for atherosclerosis. These may include adopting a healthy diet, increasing physical activity, quitting smoking, and managing underlying medical conditions such as high cholesterol, diabetes, and high blood pressure [25]. Medications may also be prescribed to address risk factors and prevent complications. Examples include statins to lower cholesterol, antiplatelet drugs such as aspirin to reduce the risk of blood clots, and blood pressure medications [26].

Hydrogen sulfide (H_2_S) is an emerging gasotransmitter with potential therapeutic applications in atherosclerosis [27]. H_2_S is produced endogenously in the body by several enzymes, including cystathionine β-synthase (CBS), cystathionine γ-lyase (CSE), and 3-mercaptopyruvate sulfurtransferase (3-MST) [28]. Recent studies have demonstrated that H_2_S plays a crucial role in regulating various aspects of cardiovascular health, including blood pressure, vascular tone, inflammation, and oxidative stress [29]. H_2_S has been shown to promote vasodilation by activating potassium channels and inhibiting calcium channels in vascular smooth muscle cells [30]. Moreover, H_2_S has been found to regulate endothelial function by stimulating the release of nitric oxide (NO), a potent vasodilator that plays a key role in maintaining vascular homeostasis. In addition, H_2_S modulates the inflammatory response by inhibiting the activation of nuclear factor kappa B (NF-κB) and promoting the production of anti-inflammatory cytokines [31]. H_2_S can also protect against oxidative stress by eliminating reactive oxygen species (ROS) and increasing the expression of antioxidant enzymes [32], such as glutathione peroxidase (GPx) and superoxide dismutase (SOD) [33].

Interestingly, H_2_S has been found to interact with other gasotransmitters, such as nitric oxide (NO), to regulate vascular function. Specifically, H_2_S has been shown to enhance NO bioavailability by inhibiting the production of reactive oxygen species (ROS) and reducing the activity of arginase, an enzyme that competes with nitric oxide synthase (NOS) for the substrate arginine. The interaction between H_2_S and NO is particularly relevant in atherosclerosis, where impaired NO bioavailability is a hallmark of endothelial dysfunction and contributes to the development and progression of the disease [34]. All these effects make H_2_S an attractive target for developing novel therapies for cardiovascular diseases, including atherosclerosis [35]. Several H_2_S-based treatments have been proposed to treat and prevent atherosclerosis, including H_2_S-releasing molecules, H_2_S donors, and modulation of endogenous H_2_S production. These therapies have shown promising results in preclinical studies, and clinical trials are currently underway to evaluate their efficacy and safety in humans. The regulation of H_2_S and oxygen homeostasis is a complex and dynamic process, and further research is needed to understand the molecular mechanisms involved and to develop safe and effective therapeutic interventions [36].

## 2. Methods

For this comprehensive review on H_2_S and oxygen homeostasis in atherosclerosis, a dedicated protocol was used:Search strategy: For research context, the search-specific syntaxes used were based on the keywords “atherosclerosis” AND “hydrogen sulfide” OR “oxygen homeostasis”. The narrative synthesis was based on a Google search, while the systematic literature review was performed using reputable international medical databases, including Elsevier, Nature, and Web of Science, articles being published between 2018 and 2023, and the selection process was based on the Preferred Reporting Items for Systematic Reviews and Meta-Analyses (PRISMA) guidelines (Table 1 and Figure 1).Article evaluation: An algorithm was developed to evaluate each article’s scientific impact and quality. The Cochrane Risk of Bias Tool, AMSTAR Checklist, and JBI Critical Appraisal Checklist were used to ensure qualitative support and data validation. In addition, the algorithm considered several factors, such as the year of publication, the total number of citations, and the PEDro score. Using this algorithm, the articles were assessed and ranked according to their scientific merit and relevance to the field. The selected papers were inputs for a benchmarking analysis strategy, and the “paradigm funnel” was used to obtain relevant reports.Data extraction: Data were extracted using a standardized form. The information collected included study design, sample size, population characteristics, intervention or exposure intervention, outcome measures, and results.Data analysis and synthesis: A narrative synthesis was performed to summarize the findings of the selected studies. The studies were organized based on their study design, population characteristics, intervention or exposure intervention, outcome measures, and results. Data were analyzed descriptively to identify common themes and patterns.Graphical techniques and tools: Data collection sheets, graphic representations, critical examination matrices, cause–effect diagrams, Pareto charts, control sheets, and correlation diagrams were used to analyze, interpret, and present the obtained data and to elaborate this comprehensive review article.

## 3. Results and Discussion

### 3.1. Oxygen Homeostasis in Atherosclerosis and the Role of Hydrogen Sulfide

Oxygen homeostasis plays a crucial role in maintaining cardiovascular health [37]. Adequate oxygen supply to the heart is necessary for proper functioning [38], and any imbalance in oxygen demand and supply can lead to cardiovascular disease [34]. Hypoxia, or low oxygen levels, can cause inflammation and oxidative stress, leading to atherosclerosis and plaque buildup in arteries, ultimately leading to heart attack and stroke [39]. On the other hand, hyperoxia, or excess oxygen levels, can also be detrimental, as it can lead to oxidative stress and tissue damage. Therefore, maintaining proper oxygen homeostasis is essential for preventing and treating cardiovascular disease [40].

Emerging evidence suggests that disruption of oxygen homeostasis is a critical factor in the development and progression of atherosclerosis [41]. Oxidative stress resulting from an imbalance between the production of reactive oxygen species (ROS) and the antioxidant defense system is a key factor that leads to endothelial dysfunction and promotes atherosclerotic plaque formation [42]. Moreover, hypoxia, a condition with insufficient oxygen supply to tissues, can foster the development of atherosclerosis by inducing metabolic dysfunction and inflammation in the arterial wall [9]. In addition, impaired oxygen diffusion and consumption in atherosclerotic plaques can lead to local hypoxia, exacerbating the inflammatory response and contributing to plaque instability and rupture. Therefore, maintaining oxygen homeostasis is critical for preventing and managing atherosclerosis [43].

ROS are essential in maintaining the signaling pathways that control cellular processes such as inflammation, differentiation, proliferation, and apoptosis [44]. However, dysregulated ROS production can lead to overstimulation of these pathways, promoting ROS-associated disease phenotypes, such as type II diabetes and atherosclerosis [45]. ROS can be generated by several enzyme systems found throughout the vascular system, including the mitochondrial electron transport chain, NADPH oxidases, xanthine oxidase, and endothelial nitric oxide synthase [46].

Mitochondria, the major cellular energy generators, play vital roles in calcium and iron homeostasis, regulating critical processes, including ROS generation, inflammation, and apoptosis [47]. In addition, mitochondrial ROS (mtROS) are essential for regulating cellular responses, such as gene expression, signal transduction, and responses to stress. The two major forms of mtROS are superoxide and hydrogen peroxide, formed when electrons “spill” onto oxygen from mitochondrial proteins located earlier in the electron transport chain [40].

Oxidative stress can also affect the function of vascular smooth muscle cells and contribute to the development of atherosclerotic plaques [48]. These cells play a role in regulating vascular tone and remodeling, and their proliferation and migration can contribute to the development of atherosclerosis [49]. Oxidative stress can promote the proliferation and migration of vascular smooth muscle cells, leading to atherosclerotic lesions [50].

Several factors can contribute to oxidative stress in atherosclerosis, including high LDL cholesterol levels, hypertension, diabetes, smoking, and aging. These factors can increase ROS production and decrease the body’s antioxidant defense mechanisms [51]. Antioxidants, such as vitamins C and E, and plant-derived compounds, such as polyphenols, have been shown to reduce oxidative stress and improve endothelial function. Other potential targets include enzymes involved in ROS production, such as NADPH oxidases, and transcription factors involved in oxidative stress responses, such as nuclear factor erythroid 2-related factor 2 (Nrf2) [52].

H_2_S has been shown to play a crucial role in regulating oxygen homeostasis in various pathophysiological conditions, including atherosclerosis [53]. H_2_S has been found to promote angiogenesis and vasodilation, leading to increased oxygen delivery to the tissues [54]. It also regulates the production of reactive oxygen species (ROS), which are known to contribute to oxidative stress and endothelial dysfunction, two key processes in the development and progression of atherosclerosis [55]. H_2_S has been shown to scavenge ROS and upregulate antioxidant enzymes, such as superoxide dismutase (SOD) and glutathione peroxidase (GPx), protecting against oxidative stress [56]. Additionally, H_2_S can inhibit the activation of nuclear factor-kappa B (NF-κB) and other proinflammatory signaling pathways, reducing the inflammatory response and promoting the resolution of inflammation. These anti-inflammatory effects of H_2_S are also thought to contribute to regulating oxygen homeostasis in atherosclerosis by reducing inflammation-mediated endothelial dysfunction and improving tissue oxygenation [57].

H_2_S can regulate oxygen-sensing enzymes and transcription factors in vascular cells through multiple molecular mechanisms [28]. One of the primary mechanisms by which H_2_S regulates oxygen sensing is through the modulation of prolyl hydroxylase domain (PHD) enzymes, which play a critical role in regulating the activity of hypoxia-inducible factors (HIFs) [58]. HIF-1α activation can promote gene expression in cell survival, angiogenesis, and glycolysis, mitigating hyperoxia’s effects [59]. Under normoxic conditions, PHD enzymes hydroxylate specific proline residues on HIFs, marking them for degradation via the von Hippel–Lindau protein-mediated ubiquitination pathway [60]. However, H_2_S can inhibit the activity of PHD enzymes, stabilizing HIFs and promoting their transcriptional activity. This leads to the upregulation of genes involved in angiogenesis, erythropoiesis, and glycolysis, which help to improve tissue oxygenation under hypoxic conditions [59]. H_2_S has been shown to have a protective role in hypoxia-induced inflammation and oxidative stress, which can contribute to the prevention of atherosclerosis [61]. In hypoxic conditions, H_2_S can activate hypoxia-inducible factor 1 alpha (HIF-1α), a transcription factor that regulates gene expression in oxygen homeostasis and cellular adaptation to hypoxia [62]. Activation of HIF-1α by H_2_S can result in the upregulation of antioxidant enzymes and cytoprotective genes, reducing oxidative stress and inflammation in vascular cells [63].

In addition to regulating PHD enzymes, H_2_S can also directly modulate the activity of HIFs by promoting the expression of the HIF-1α inhibitor, factor-inhibiting HIF (FIH). FIH hydroxylates an asparagine residue on HIF-1α, inhibiting its transcriptional activity. Therefore, H_2_S can upregulate FIH expression, reducing the transcriptional activity of HIF-1α and preventing excessive angiogenesis and vascular leakage [64].

H_2_S has also been shown to be protective against hyperoxia-induced injury in vascular cells. Studies have demonstrated that H_2_S can alleviate oxidative stress in hyperoxic conditions by upregulating antioxidant enzymes and reducing reactive oxygen species (ROS) production. In addition, H_2_S can modulate the expression and activity of hypoxia-inducible factor 1α (HIF-1α), a key regulator of oxygen homeostasis, in response to hyperoxia [65].

### 3.2. Inflammation in Atherosclerosis and the Role of Hydrogen Sulfide

The interplay between oxidative stress and inflammation is connected to the pathogenesis of atherosclerosis, promoting endothelial dysfunction, foam cell formation, and plaque progression [66]. The chronic inflammation present in the vessel wall initiates and perpetuates atherosclerotic lesion formation [67]. Inflammatory cytokines, chemokines, and adhesion molecules promote the adhesion and recruitment of leukocytes, such as monocytes and T lymphocytes, to the site of inflammation [68]. Once recruited, these immune cells produce reactive oxygen species (ROS) and proinflammatory mediators, further exacerbating the inflammatory response and causing endothelial dysfunction [69].

ROS generated by endothelial cells, smooth muscle cells, and immune cells can oxidize lipids, including LDL, forming oxidized LDL (ox-LDL), which triggers the production of proinflammatory cytokines, chemokines, and adhesion molecules, as well as the expression of scavenger receptors on macrophages, promoting the uptake of ox-LDL by these cells and the formation of foam cells. Foam and other inflammatory cells contribute to the construction of atherosclerotic plaque [70]. Moreover, oxidative stress also impairs the function of endothelial nitric oxide synthase (eNOS), decreasing the production of nitric oxide (NO), a potent vasodilator and anti-inflammatory molecule [71]. Reduced NO bioavailability leads to endothelial dysfunction, promoting the progression of atherosclerosis [72].

H_2_S has been reported to possess potent anti-inflammatory and antioxidant properties, which could be beneficial in preventing and treating atherosclerosis. One of the primary targets of H_2_S in regulating inflammation is the NF-κB pathway. H_2_S has been shown to inhibit the activation of NF-κB in endothelial cells and macrophages by suppressing the phosphorylation and degradation of IκBα, an inhibitor of NF-κB. This inhibition leads to reduced production of proinflammatory cytokines, such as IL-6, IL-1β, and TNF-α [73]. In addition, H_2_S has been shown to enhance the expression and activity of nuclear factor erythroid 2-related factor 2 (Nrf2), a transcription factor that regulates the expression of antioxidant and cytoprotective genes [74]. H_2_S can activate Nrf2 by promoting its nuclear translocation and binding to antioxidant response elements (ARE) in the promoter region of target genes [75].

H_2_S also exerts anti-inflammatory effects by modulating the expression of adhesion molecules, such as intercellular adhesion molecule-1 (ICAM-1) and vascular cell adhesion molecule-1 (VCAM-1) in endothelial cells. H_2_S can downregulate the expression of these molecules and prevent the adhesion and infiltration of inflammatory cells into the vessel wall [76]. Furthermore, H_2_S can modulate the function of immune cells involved in atherosclerosis. H_2_S can promote the differentiation of regulatory T-cells (Tregs), suppressing proinflammatory T-cell activation. H_2_S can also inhibit the proliferation and activation of macrophages, reducing the uptake of oxidized low-density lipoprotein (ox-LDL) and foam cell formation [77].

### 3.3. Results Seen as Progress in the Last 5 Years (2018–2023) Resulting from PRISMA-Type Systematic Review

Investigating the mechanisms underlying the protective effects of H_2_S on adipose tissue inflammation and atherosclerosis development, including the regulation of oxidative stress, autophagy, and immune responses, and whether H_2_S is involved in the browning of white adipocytes is also a critical topic regarding H_2_S as a double-edged sword in vivo; a high dose has a toxic effect, while a low dose has a therapeutic effect. Defining the therapeutic window is fundamental for developing H_2_S medicinal drugs [78,79].

H_2_S, found to exhibit a range of physiological functions in maintaining vascular homeostasis, has also been shown to suppress proatherogenic cellular responses in the progression of atherosclerosis. Hemoglobin (Hb) oxidation is also a critical factor in atherosclerosis development, and the presence of Hb in atherosclerotic lesions can contribute to plaque lipids’ oxidation and vascular inflammation. However, H_2_S has been shown to have protective effects on vascular plaque development provoked by Hb and heme by modifying lipid oxidation and vascular inflammation [80].

Protein kinase C (PKC) and Akt are essential signal transduction molecules in cells, and their activation levels change during atherosclerosis development and progression. H_2_S levels were lower in patients with maintenance hemodialysis compared with the normal population, and this was associated with increased cPKCβII activation and decreased Akt phosphorylation. H_2_S has a synergistic effect with nitric oxide (NO) on vasodilation. H_2_S can regulate the production of eNOS through a specific pathway and participate in the regulation of endothelial function [81].

In the last years, research on H_2_S has identified several new mechanisms by which it exerts vasorelaxant effects, including activation of potassium and calcium channels, inhibiting adenylyl cyclase, and stimulating phosphodiesterases. H_2_S also forms polysulfides with NO to activate transient receptor potential ankyrin 1 (TRPA1) channels in vasodilation, and H_2_S -NO hybrid molecules have been shown to have better vasorelaxant and angiogenesis activity. Regarding post-translational modification, H_2_S can sulfhydrate Kelch-like ECH-associated protein 1 (KEAP1) to dissociate with Nrf2 and enhance antioxidant responses, leading to amelioration of diabetes-accelerated atherosclerotic progress. Furthermore, recent studies suggest that H_2_S may improve atherosclerosis by activating SIRT1, which has been implicated in regulating vascular tone and preventing endothelial senescence [82].

One of the novelties in the last years is the identification of CSE-derived H_2_S as an endogenous inhibitor of human antigen R (HuR) and atherosclerotic vascular disease progression. This discovery is a new function for H_2_S. Furthermore, sulfhydration of HuR is required to lower the expression of target mRNAs, such as E-selectin, by preventing binding and subsequent mRNA stabilization. Genetic ablation of CSE in endothelial cells is sufficient to increase E-selectin expression and accelerate atherosclerosis [83].

Recent studies have shown that abnormal activation of platelets contributes to atherothrombosis, and the recruitment of leukocytes to the thrombus plays an essential role in the stability of the thrombus. H_2_S has been found to inhibit platelet activation and aggregation. S-sulfhydration has been identified as a major mechanism of the physiological effects of H_2_S. GYY4137, a water-soluble, long-acting H_2_S donor chemical, is gaining popularity among researchers due to its ability to compensate for unstable H_2_S emissions during the hydrolysis of inorganic sulfates [84].

Recent analyses have revealed that the synthesis of H_2_S is progressively decreased during the development of atherosclerosis in fat-fed apoE2/2 mice, which may indicate the potential use of H_2_S donor drugs in treating atherosclerosis. Experiments with NaHS, an H_2_S donor, and a CSE inhibitor support this claim by showing that exogenous H_2_S supplementation can prevent or reduce plaque formation and that endogenous H_2_S depletion can aggravate atherosclerotic lesions. Additionally, H_2_S has been found to regulate the expression of chemokines and their receptors, particularly CX3CL1–CX3CR1, in stimulated macrophages [85].

H_2_S can reduce atherosclerosis caused by hyperhomocysteinemia (HHcy) by reducing endothelial ER stress. HHcy-induced persistent endoplasmic reticulum (ER) stress leads to caspase-12-dependent endothelial apoptosis and exacerbates atherosclerosis development. H_2_S exhibits antiatherogenic effects by sulfhydrating protein disulfide isomerase (PDI) to increase its activity, which reduces HHcy-induced endothelial ER stress [86].

H_2_S has been shown to play a role in atherosclerosis by inhibiting the activity of the matrix metalloproteinases MMP-2, MMP-9, and MMP-8. Studies have shown that H_2_S S inhibits MMP-2 and MMP-9 activity, which are involved in neointima formation created by smooth muscle cell migration from the media to the intima. This inhibitory effect of H_2_S on MMP-2 and MMP-9 activity reduces the development of atherosclerotic lesions. Furthermore, H_2_S has also been shown to inhibit the activity of MMP-8, which is involved in atherosclerotic lesion development. MMP-8 plays a significant role in the proteolytic activity on matrix proteins and fibrillar collagens, as well as nonmatrix proteins, such as angiotensin I. Inhibiting MMP-8 activity through the action of H_2_S can prevent complications in the degradation of these proteins, which can contribute to the formation and progression of atherosclerosis. Therefore, the role of H_2_S in plaque alleviation is primarily through its inhibitory effects on MMP-2, MMP-9, and MMP-8, which are all involved in the development and progression of atherosclerosis [87].

H_2_S can be synthesized through enzymatic and nonenzymatic pathways and is produced abundantly by the gut microbiota, particularly by sulfate-reducing bacteria. The concentration of H_2_S in the intestine is higher than in other body parts. Changes in the gut microbiota and the integrity of the gut–blood barrier may have implications for the progression of atherosclerosis, as well as other inflammatory diseases [88,89].

Recent studies have shown that H_2_S can modulate HIF signaling in various cell types. H_2_S has been shown to reduce HIF1α activity in endothelial cells, leading to decreased angiogenesis and improved vascular function. In addition, H_2_S has been shown to have immunomodulatory effects and can modulate immune cell phenotype, function, and metabolism. H_2_S has been shown to enhance glycolysis in T cells, leading to increased cytokine production and immune cell activation. H_2_S has also been shown to promote the differentiation of regulatory T cells, which play a key role in immune tolerance and preventing autoimmune diseases. H_2_S has also been shown to modulate the function of immune cells in the gut, leading to improved gut barrier function and the prevention of inflammatory bowel disease. Overall, the modulation of HIF signaling by H_2_S appears to be an essential mechanism by which H_2_S can regulate various physiological and pathological processes, including metabolism and immune function. Further studies are needed to fully understand the mechanisms underlying the effects of H_2_S on HIF signaling and the potential therapeutic applications of H_2_S in various diseases [59].

Recent studies have demonstrated a potential relationship between the decreased level of H_2_S and the pathogenesis of atherosclerosis. H_2_S has been found to suppress the function of a thioredoxin-interacting protein (TXNIP), which is responsible for the excessive production of inflammatory cytokines such as IL18 and IL1β via the activation of NOD-like receptor family pyrin domain-containing protein3 (NLRP3) inflammasome. In addition, the activation of inflammasome in macrophages or endothelial cells has been suggested to be an essential step in the development of atherosclerosis [90].

Different types of synthetic organic H_2_S donors and H_2_S-releasing molecules have been reported. However, the pathogenesis of atherosclerosis is complex, and only a few synthetic organic H_2_S donors have demonstrated therapeutic potential for antiatherosclerosis. The safety and effectiveness of H_2_S donors are key issues that need to be considered [91].

### 3.4. Main Outlines, Limitations, and Future Directions

The publications included in the review cover a wide range of topics related to atherosclerosis, oxidative stress, inflammation, and vascular disease. The main topics covered by these publications include the pathophysiology of atherosclerosis, the role of oxidative stress and inflammation in the development of atherosclerosis, the impact of lifestyle factors on cardiovascular health, the therapeutic potential of hydrogen sulfide in cardiovascular disease, and the molecular mechanisms underlying vascular disease.

The main issues addressed in these publications include identifying new therapeutic targets for atherosclerosis, the role of oxidative stress and inflammation in cardiovascular disease, the importance of lifestyle interventions for preventing cardiovascular disease, and the potential of hydrogen sulfide as a therapeutic agent for cardiovascular disease.

The limitations of these publications include that many of them are based on animal studies or in vitro experiments, which may not accurately reflect the complexity of human disease. Additionally, many of these publications focus on specific molecular mechanisms or pathways, which may not provide a complete picture of the underlying pathophysiology of atherosclerosis and cardiovascular disease.

H_2_S has been shown to have various effects on vascular smooth muscle cells, endothelial cells, and macrophages that may be relevant to atherosclerosis [84]. In addition to its impact on inflammation, oxidative stress, and oxygen homeostasis, H_2_S has also been shown to regulate various cellular processes, such as cell proliferation, migration, apoptosis, and autophagy, which are all involved in the pathogenesis of atherosclerosis [29].

In vascular smooth muscle cells (VSMCs), H_2_S has been shown to induce relaxation and inhibit proliferation, migration, and calcification [92]. H_2_S can activate the KATP channel, which leads to membrane hyperpolarization and decreased Ca^2+^ influx, ultimately leading to vasodilation [93]. H_2_S can also inhibit VSMC proliferation and migration by inhibiting the MAPK pathway and downregulating cyclin D1 expression [94]. Additionally, H_2_S has been shown to inhibit VSMC calcification by suppressing the Wnt/beta-catenin pathway [95]. Studies have demonstrated that H_2_S can regulate cell proliferation in VSMCs by inhibiting cell cycle progression and inducing cell cycle arrest by downregulating the expression of cyclin D1 and CDK4, critical regulators of the G1 phase [96]. Additionally, H_2_S has been shown to induce apoptosis in VSMCs by activating the caspase-dependent apoptotic pathway [97].

In endothelial cells, H_2_S has been shown to promote angiogenesis and inhibit inflammation and oxidative stress [98]. H_2_S can activate endothelial nitric oxide synthase (eNOS), producing more NO, a potent vasodilator and anti-inflammatory molecule [99]. H_2_S can also inhibit oxidative stress in endothelial cells by upregulating antioxidant enzymes, such as catalase and superoxide dismutase [100]. In addition, H_2_S has been shown to promote angiogenesis by inducing the expression of vascular endothelial growth factor (VEGF) [101]. H_2_S promotes angiogenesis by activating the Akt and ERK signaling pathways, which are involved in the regulation of cell migration and angiogenesis [102].

In macrophages, H_2_S has been shown to inhibit inflammation and promote cholesterol efflux [103]. H_2_S can inhibit the production of proinflammatory cytokines, such as interleukin-1 beta (IL-1β) and tumor necrosis factor-alpha (TNF-α), in macrophages [104]. H_2_S can also promote cholesterol efflux from macrophages by upregulating the expression of ATP-binding cassette transporter A1 (ABCA1) and scavenger receptor class B type I (SR-BI), key proteins involved in cholesterol efflux from macrophages [105]. Moreover, H_2_S has been shown to regulate cell apoptosis and autophagy in macrophages. H_2_S has been shown to induce macrophage apoptosis by activating the caspase-dependent apoptotic pathway [106]. Additionally, H_2_S has been shown to regulate macrophage autophagy by promoting the expression of autophagy-related genes and increasing the formation of autophagosomes [107].

H_2_S can interact with several signaling pathways involved in the pathogenesis of atherosclerosis, including the PI3K/Akt, MAPK/ERK, and NF-κB pathways [31]. The PI3K/Akt pathway plays a critical role in cell survival, proliferation, and apoptosis. H2S has been shown to activate this pathway in vascular smooth muscle cells (VSMCs), leading to increased cell survival and proliferation [108]. Additionally, H_2_S has been shown to increase Akt phosphorylation in endothelial cells, leading to enhanced angiogenesis [109].

The MAPK/ERK pathway is involved in cell proliferation, differentiation, and survival. H_2_S has been shown to activate this pathway in VSMCs, leading to increased cell proliferation and migration. Additionally, H_2_S has been shown to activate the ERK pathway in endothelial cells, leading to enhanced angiogenesis [110].

H_2_S has been shown to modulate the activity of endothelial nitric oxide synthase (eNOS) in vascular cells. eNOS is an enzyme that plays a crucial role in regulating vascular function by producing NO, a potent vasodilator [111]. Several mechanisms are proposed for the modulation of eNOS activity by H_2_S. One mechanism involves the direct binding of H_2_S to eNOS, leading to the formation of a thiol-activated form of the enzyme [112]. H_2_S has also been shown to stimulate eNOS activity by enhancing the availability of its cofactor, tetrahydrobiopterin (BH4), which is critical for eNOS activity [113]. In addition to its direct effects on eNOS, H_2_S has been shown to modulate eNOS activity by regulating signaling pathways involved in eNOS activation. H_2_S has been reported to activate the PI3K/Akt pathway, which leads to the phosphorylation and activation of eNOS [114]. H_2_S has also been shown to activate the AMP-activated protein kinase (AMPK) pathway, which plays a critical role in regulating eNOS activity [115].

Several approaches have been explored to increase H_2_S levels in the body, including H_2_S donors, H_2_S-releasing agents, and inhibitors of H_2_S-degrading enzymes. Organic molecules, such as sodium hydrosulfide (NaHS) and sodium sulfide (Na_2_S), have been used as H_2_S donors. These molecules can release H_2_S in a controlled manner, leading to a sustained increase in H_2_S levels [116]. Some drugs have been designed to release H_2_S in the body as a secondary effect. For example, the anti-inflammatory drug naproxen has been modified to include an H_2_S-releasing moiety, resulting in a dual therapeutic effect [117]. Enzymes such as cystathionine γ-lyase (CSE) and 3-mercaptopyruvate sulfurtransferase (3-MST) are involved in the degradation of H_2_S in the body. Inhibitors of these enzymes, such as DL-propargylglycine (PAG) and β-cyano-l-alanine (BCA), can increase H_2_S levels by preventing their breakdown [118].

Clinical studies on H_2_S-based therapies for atherosclerosis are still in the early stages. However, preclinical studies have shown promising results, suggesting that H_2_S-based therapies could be a potential treatment strategy for atherosclerosis (Figure 2). H_2_S-based therapies have several advantages, including multitargeted effects, and low cost. H_2_S can be delivered through various noninvasive routes, including inhalation, injection, ingestion. H_2_S-based therapies are relatively low-cost compared with many other treatments for atherosclerosis [119]. However, some limitations to H_2_S-based therapies include a lack of standardized dosing, the short half-life of H_2_S, and potential side effects, such as hypotension and respiratory depression. The development of H_2_S-based therapies for atherosclerosis faces several challenges and requires further research to overcome these obstacles. Some of the future directions include:➢Delivery: The effective delivery of H_2_S to the target site is crucial for its therapeutic potential. Researchers need to develop suitable delivery methods that ensure adequate and sustained H_2_S release at the site of action.➢Dose optimization: The optimal dose of H_2_S required for therapeutic benefit must be determined. The dose must be sufficient to produce the desired effects without causing adverse effects.➢Safety: The safety of H_2_S-based therapies needs to be established. Researchers need to determine the potentially toxic effects of H_2_S and establish safe dosage ranges.➢Combination therapy: H_2_S-based therapies can be combined with other therapies, such as statins or anti-inflammatory drugs, to enhance their effectiveness [120].➢Personalized medicine: H_2_S-based therapies may not be effective for all patients with atherosclerosis. Personalized medicine approaches can be used to identify patients who are likely to respond to H_2_S-based treatments.➢Clinical trials: Trials are needed to determine the safety and efficacy of H_2_S-based therapies in humans.

### 3.5. Meta-Analysis

PICOT analysis [121] related to atherosclerosis:Population: Adults (aged 18 years or older) diagnosed with atherosclerosis:Intervention: Drug and placebo;Comparison: Placebo or standard treatment (e.g., statins)’;Outcome: Reduction in cardiovascular events (e.g., myocardial infarction, stroke, death);Time frame: Last 10 years.

The meta-analysis analyzed the data from several clinical trials, investigating the effectiveness of different drugs compared with placebo in reducing cardiovascular events in patients with atherosclerosis over the past 10 years. Table 2 lists the studies selected after searching at https://clinicaltrials.gov/ (accessed on 1 April 2023), as well as the number of participants enrolled, intervention/study design, and outcome measures/observations. Figure 3 summarizes the adverse effects of various drugs versus placebo in atherosclerosis. From the initial eight trials listed in Table 2, only six qualified for meta-analysis, providing the same outcome measure. The meta-analysis provides a lack of bias for the included studies and slightly unbalanced adverse effects reported for drugs compared with placebo for atherosclerosis. The meta-analysis emphasizes the need for further trials on this subject.

The numerical values of H_2_S content are provided in a single observational study with NCT number NCT01407172, which enrolled 252 participants with peripheral arterial disease. Unfortunately, no other studies have reported similar data on H_2_S content in patients with atherosclerosis.

## 4. Conclusions

Atherosclerosis is a complex multifactorial disease involving inflammation, oxidative stress, and dysregulation of oxygen homeostasis. H_2_S, a gasotransmitter, has emerged as a promising therapeutic target for atherosclerosis due to its anti-inflammatory, antioxidant, and proangiogenic properties. H_2_S can modulate multiple signaling pathways in atherosclerosis, including regulating oxygen-sensing enzymes, inflammation, oxidative stress, and effects on cell proliferation, migration, apoptosis, and autophagy. However, H_2_S-based therapies have shown promise in preclinical studies. Performed meta-analysis regarding the adverse effects of various drugs versus placebo in atherosclerosis indicates that adverse events are greater than those in the placebo group. The funnel plot concludes with no biased studies.

## Figures and Tables

**Figure 1 ijms-24-08376-f001:**
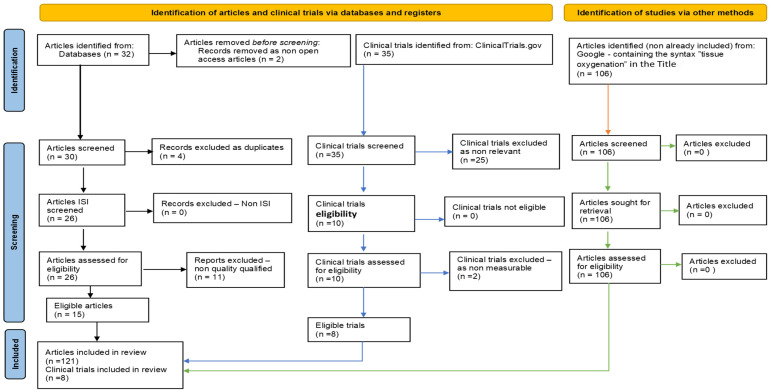
Flow diagram (PRISMA type), customized for our study.

**Figure 2 ijms-24-08376-f002:**
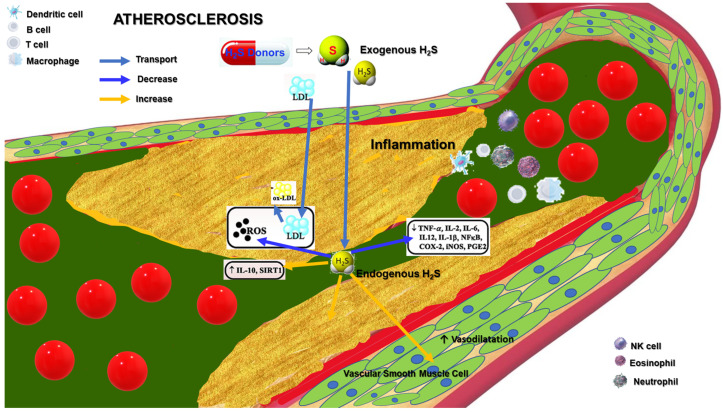
The complex interplay between H_2_S effects and tissue oxygenation homeostasis in atherosclerosis.

**Figure 3 ijms-24-08376-f003:**
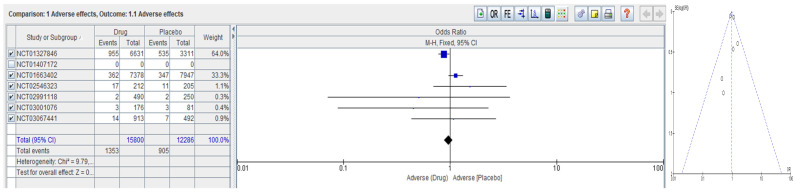
Meta-analysis regarding the adverse effects of various drugs versus placebo in atherosclerosis.

**Table 1 ijms-24-08376-t001:** Keyword sets-based search for related articles—numerical results in specific international scientific databases.

Keywords	Elsevier	Nature	Web of Science	Total
“atherosclerosis” AND “hydrogen sulfide” OR “oxygen homeostasis”	5	13	12	30
Articles after duplicates exclusion				26
Articles after nonrelevant exclusion				15

**Table 2 ijms-24-08376-t002:** Studies selected after searching at https://clinicaltrials.gov/ (accessed on 1 April 2023).

No.	NCT No.	Study Title	Interventions/Study Design	Number Enrolled	Outcome Measures/OBS.
1	NCT01407172	Hydrogen Sulfide and Peripheral Arterial Disease [122]	Observational	252	Plasma free H_2_S, nmol/L: No Vasc Disease, n = 53: 368.53 ± 20.8; Vasc Disease, n = 140: 441.04 ± 15.40 (*p* = 0.010); PAD, n = 13: 514.48 ± 62.05 (*p* = 0.007)
3	NCT02546323	A Phase 3 Study Measuring the Effect of Rosuvastatin 20 mg on Carotid Intima–Media Thickness in Chinese Subjects With Subclinical Atherosclerosis	Drug: Rosuvastatin Drug: Placebo	543	Percent in Lipid, Lipoprotein, and Apolipoprotein
4	NCT01327846	Cardiovascular Risk Reduction Study (Reduction in Recurrent Major CV Disease Events)	Drug: Canakinumab Drug: Placebo	10,066	Adverse Events
5	NCT01663402	ODYSSEY Outcomes: Evaluation of Cardiovascular Outcomes after an Acute Coronary Syndrome during Treatment with Alirocumab	Drug: Alirocumab Drug: Placebo	18,924	Adverse Events
6	NCT03001076	Evaluation of the Efficacy and Safety of Bempedoic Acid (ETC-1002) as Add-on to Ezetimibe Therapy in Patients with Elevated LDL-C (CLEAR Tranquility)	Drug: Bempedoic acid Drug: Ezetimibe Other: Placebo	269	Percent in Lipid, Lipoprotein, and Apolipoprotein Adverse Events
7	NCT03067441	Assessment of the Long-Term Safety and Efficacy of Bempedoic Acid (CLEAR Harmony OLE)	Drug: Bempedoic acid	1462	Percent in Lipid, Lipoprotein, and Apolipoprotein
8	NCT02991118	Evaluation of Long-Term Efficacy of Bempedoic Acid (ETC-1002) in Patients with Hyperlipidemia at High Cardiovascular Risk	Drug: Bempedoic acid Drug: placebo	779	Percent in Lipid, Lipoprotein, and Apolipoprotein

## Data Availability

Not applicable.

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
