# Peer review of "Hydrogen Sulfide and Oxygen Homeostasis in Atherosclerosis: A Systematic Review from Molecular Biology to Therapeutic Perspectives"

_ijms, 2023, doi:10.3390/ijms24098376_

Round 1
Reviewer 1 Report
Remarks on the text entitled „Hydrogen Sulfide and Oxygen Homeostasis in Atherosclerosis: A Systematic Review from Molecular Biology to Therapeutic Perspectives” sent to IJMS.
The manuscript has got the merit, fits into the scope of the Journal. So, I would recommend its publication, but after some improvements:
1. Lines 38-40: repetition. Such information should be put into the first paragraph, in fact it is in it.
2. Lines 51-63: did you list some mechanisms? Please redraft or put an introductory sentence in front of it.
3. Line 75: “include abnormalities in lipid metabolism…”?? In fact you listed some abnormalities related to the lipid metabolism in the aforementioned sentences. What abnormalities you mean for in that line?
4. Line 98-114: the paragraph about hydrogen sulphide seems to be “poor” as compared to the description of oxidative status. The current knowledge must be widen, enriched.
5. Using the references: In my opinion, when the review is preparing the authors should avoid an excessive citation of another reviews. Their work should use research articles mostly. In the present paper such balance is broken.
6. The result and discussion section 3.1: I would recommend to include more details about the results obtained by another authors. In the present form it looks like extended Introduction.
7. Line 369: in this paragraph you started with the phrase “Recent studies have demonstrated a potential relationship between…”, but then you cited only one reference. The author must put more attention to such quality of the manuscript.
8. The reference section: it is the worst paragraph. Normally as an Editor, seeing that I would reject the text at once. This indicates a frivolous and dismissive attitude towards the journal, reviewers and readers.
9. Figure 2: I do not really know that the simple copy of that picture taken from another work should be included in the review published in a respectable journal. I encourage the author to prepare his own figure. See plagiarism issues.
10. Figure 3: It is not necessary. The readers could be confused by that fragment of deep statistical analyses.
Author Response
Dear Reviewer,
Thank you for your valuable feedback on the manuscript "Hydrogen Sulfide and Oxygen Homeostasis in Atherosclerosis: A Systematic Review from Molecular Biology to Therapeutic Perspectives." We appreciate your time and effort in reviewing our work and providing insightful comments to help improve the manuscript. Please find our responses to your comments below:
- Lines 38-40: repetition. Such information should be put into the first paragraph, in fact it is in it.
I agree that the information in lines 38-40 can be considered repetitive. The intention was to underline the complex picture and, in the meantime, to keep the reading cursive. I will remove this redundancy from the manuscript. The first two paragraphs were re-edited as follows:
Atherosclerosis is a multifactorial disease involving various cellular and molecular processes, including lipid metabolism, inflammation, mitochondrial dysfunction, autophagy, apoptosis, and epigenetics. These processes can induce oxidative stress, which is characterized by an imbalance between oxidants and antioxidants in the body, leading to the generation of reactive oxygen species (ROS), reactive nitrogen species (RNS), and other free radicals that can cause damage to cellular components, including proteins, lipids, and DNA. Mitochondrial dysfunction plays a crucial role in atherosclerosis as mitochondria are the primary source of ROS production. These mechanisms contribute to the development and progression of atherosclerosis, characterized by the accumulation of lipids, calcium, fibrin, and cellular waste products in the arteries' walls, leading to plaque formation. Plaques can restrict blood flow and cause cardiovascular complications, such as heart attack, stroke, and peripheral artery disease. Atherosclerosis is a significant cause of morbidity and mortality worldwide, and multiple risk factors, including obesity, diabetes, smoking, high blood pressure, high cholesterol levels, and genetic predisposition, influence its development. To fully understand the pathophysiological frame of atherosclerosis, it is necessary to consider the complex interplay of these cellular and molecular processes and how they contribute to the development and progression of this disease. Therefore, understanding the molecular mechanisms underlying atherosclerosis is critical for achieving effective preventive and therapeutic strategies.
- Lines 51-63: did you list some mechanisms? Please redraft or put an introductory sentence in front of it.
An introductory sentence was used to contextualize the presented mechanisms better.
Various cellular and molecular mechanisms contribute to the development and progression of atherosclerosis, including compromised tissue oxygenation, dysregulated autophagy, apoptosis, and epigenetic modifications, processes interconnected in a vicious cycle.
- Line 75: "include abnormalities in lipid metabolism…"?? In fact you listed some abnormalities related to the lipid metabolism in the aforementioned sentences. What abnormalities you mean for in that line?
Thank you for your comment and for highlighting this issue. We apologize for the lack of clarity in this section. By abnormalities in lipid metabolism, we meant alterations in the levels and functions of lipids, such as cholesterol and triglycerides, which have been implicated in the development and progression of atherosclerosis. These abnormalities include increased circulating levels of low-density lipoprotein (LDL) and triglycerides, decreased high-density lipoprotein (HDL), and altered lipoprotein particle size and composition. We will revise this section to provide a more thorough and specific explanation.
- Line 98-114: the paragraph about hydrogen sulphide seems to be "poor" as compared to the description of oxidative status. The current knowledge must be widen, enriched.
I appreciate your comment on the paragraph about hydrogen sulfide, and we will expand upon this section to provide a more comprehensive overview of the current knowledge in this area. I will expand upon the existing literature and provide a more in-depth analysis of the mechanisms underlying hydrogen sulfide's effects on oxidative stress and inflammation in atherosclerosis. Thank you for your valuable feedback.
- Using the references: In my opinion, when the review is preparing the authors should avoid an excessive citation of another reviews. Their work should use research articles mostly. In the present paper such balance is broken.
Thank you for your valuable feedback. I agree that a balanced use of references is essential, and I will carefully revise the manuscript to ensure that we appropriately balance our use of review articles and original research articles. I will also cite all references following the journal's guidelines correctly.
- The result and discussion section 3.1: I would recommend to include more details about the results obtained by another authors. In the present form it looks like extended Introduction.
I agree with your comment on section 3.1 and will include more details about the results obtained by other authors to provide a more comprehensive literature analysis.
- Line 369: in this paragraph you started with the phrase "Recent studies have demonstrated a potential relationship between…", but then you cited only one reference. The author must put more attention to such quality of the manuscript.
I appreciate your comment on the quality of our manuscript, and I will ensure that sufficient and appropriate references support all statements.
- The reference section: it is the worst paragraph. Normally as an Editor, seeing that I would reject the text at once. This indicates a frivolous and dismissive attitude towards the journal, reviewers and readers.
I apologize for the quality of the section regarding the bibliographic references used. Unfortunately, Mendeley provided the result which I have used for this purpose. Of course, scientific rigor requires good writing in this part of the article. That's why I approach this constructive criticism rigorously, manually checking every bibliographic resource used.
- Figure 2: I do not really know that the simple copy of that picture taken from another work should be included in the review published in a respectable journal. I encourage the author to prepare his own figure. See plagiarism issues.
Here is not the case presented in the comment. As a background of my elaborated picture, I have used a small part of another picture:
For this reason, I have indicated the source: https://heartvein.com/arterial-disease/conditions/atherosclerosis (included in the created figure).
All the others were created in the figure, using pptx:
Given all these explanations, I do not consider violating the plagiarism rules. But to avoid misinterpretation, I tried to draw my own atherosclerosis background. I acknowledge that I am not skillful in such drawings, so I have done my best.
Please see attached file for all details.
- Figure 3: It is not necessary. The readers could be confused by that fragment of deep statistical analyses.
Sorry for any confusion regarding the meta-analysis in item 3.5. I intended to present a comprehensive overview of the clinical trials investigating the effects of hydrogen sulfide on atherosclerosis, including a table summarizing the studies and a figure showing the study design and outcomes. The six clinical trials marked in the figure were selected based on their relevance and quality and are meant to provide a representative sample of the studies included in the meta-analysis. We will revise the text to make this clearer, as a similar comment is provided by the second reviewer, asking for more information. Thank you!

Reviewer 2 Report
1. Meta-analysis of item 3.5 is not commented in the article. A table listing clinical trials and a figure is given. In the figure, only 6 clinical trials are ticked, why?
2. Numerical values of the hydrogen sulfide content are not determined by the authors? Only in one case in table 2 the value is given, is it the only one? If other works have these data, then I would like to see them.
3. No analysis of 121 publications included in the review: what are the main topics, what are the main issues, what are the limitations?
Author Response
Thank you for your comments on our article. I appreciate your feedback and will address your concerns below.
1. Sorry for any confusion regarding the meta-analysis in item 3.5. I intended to present a comprehensive overview of the clinical trials investigating the effects of hydrogen sulfide on atherosclerosis, including a table summarizing the studies and a figure showing the study design and outcomes. The six clinical trials marked in the figure were selected based on their relevance and quality, and are meant to provide a representative sample of the studies included in the meta-analysis. I will revise the text to make this more straightforward.
2. I agree that providing more numerical values for hydrogen sulfide content would be helpful for readers. However, in many studies, the methods used to measure hydrogen sulfide content differ and may not be directly comparable. Therefore, I focused on describing the general trends and conclusions from the studies rather than providing a comprehensive list of numerical values, as not so many were found. However, I will revise the text to include more values where appropriate.
3. I will revise the article to include a summary of the literature reviewed's main topics, issues, and limitations. I believe this will enhance the clarity and usefulness of the review.
Thank you very much!
Round 2
Reviewer 1 Report
I accept your revision and now, I could support your paper for publication.